# Language as a Cognitive Tool to Imagine Goals in Curiosity-Driven Exploration

**Cédric Colas\*, Tristan Karch\***
Inria - Flowers team
Université de Bordeaux
{firstname.lastname}@inria.fr

**Nicolas Lair\***
Inserm U1093
Cloud Temple
nicolas.lair@inserm.fr

**Jean-Michel Dussoux**
Cloud Temple
Paris

**Clément Moulin-Frier**
Inria - Flowers team
Université de Bordeaux
ENSTA ParisTech

**Peter Ford Dominey**
Inserm U1093
Université de Dijon

**Pierre-Yves Oudeyer**
Inria - Flowers team
Université de Bordeaux
ENSTA ParisTech

## Abstract

Developmental machine learning studies how artificial agents can model the way children learn open-ended repertoires of skills. Such agents need to create and represent goals, select which ones to pursue and learn to achieve them. Recent approaches have considered goal spaces that were either fixed and hand-defined or learned using generative models of states. This limited agents to sample goals within the distribution of known effects. We argue that the ability to imagine out-of-distribution goals is key to enable creative discoveries and open-ended learning. Children do so by leveraging the compositionality of language as a tool to imagine descriptions of outcomes they never experienced before, targeting them as goals during play. We introduce IMAGINE, an intrinsically motivated deep reinforcement learning architecture that models this ability. Such imaginative agents, like children, benefit from the guidance of a social peer who provides language descriptions. To take advantage of goal imagination, agents must be able to leverage these descriptions to interpret their imagined out-of-distribution goals. This generalization is made possible by modularity: a decomposition between learned goal-achievement reward function and policy relying on deep sets, gated attention and object-centered representations. We introduce the Playground environment and study how this form of goal imagination improves generalization and exploration over agents lacking this capacity. In addition, we identify the properties of goal imagination that enable these results and study the impacts of modularity and social interactions.

## 1 Introduction

Building autonomous machines that can discover and learn open-ended skill repertoires is a long-standing goal in Artificial Intelligence. In this quest, we can draw inspiration from children development [12]. In particular, children exploration seems to be driven by intrinsically motivated brain processes that trigger spontaneous exploration for the mere purpose of experiencing novelty, surprise or learning progress [32, 42, 45]. During *exploratory play*, children can also invent and pursue their own problems [19].

---

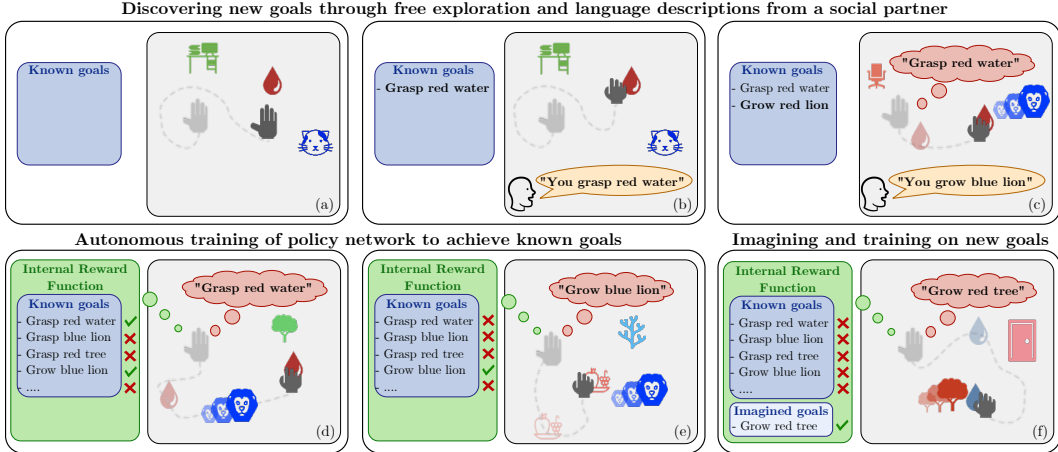

Figure 1: **IMAGINE overview**. In the *Playground* environment, the agent (hand) can move, grasp objects and grow some of them. Scenes are generated procedurally with objects of different types, colors and sizes. A social partner provides descriptive feedback (orange), that the agent converts into targetable goals (red bubbles).

Algorithmic models of intrinsic motivation were successfully used in developmental robotics [55, 6], in reinforcement learning [16, 63] and more recently in deep RL [8, 56]. Intrinsically Motivated Goal Exploration Processes (IMGEP), in particular, enable agents to sample and pursue their own goals without external rewards [7, 26, 27] and can be formulated within the deep RL framework [25, 53, 22, 58, 71, 60]. However, representing goal spaces and goal-achievement functions remains a major difficulty and often requires hand-crafted definitions. Past approaches proposed to learn image-based representations with generative models such as Variational Auto-Encoders [46, 53], but were limited to the generation of goals within the distribution of already discovered effects. Moving beyond *within-distribution* goal generation, *out-of-distribution* goal generation could power creative exploration in agents, a challenge that remains to be tackled.

In this difficult task, children leverage the properties of language to assimilate thousands of years of experience embedded in their culture, in a only a few years [67, 10]. As they discover language, their goal-driven exploration changes. Piaget [57] first identified a form of egocentric speech where children narrate their ongoing activities. Later, Vygotsky [72] realized that they were generating novel plans and goals by using the expressive generative properties of language. The harder the task, the more children used egocentric speech to plan their behavior [72, chap. 2]. Interestingly, this generative capability can push the limits of the real, as illustrated by Chomsky [18]'s famous example of a sentence that is syntactically correct but semantically original "*Colorless green ideas sleep furiously*". Language can thus be used to generate out-of-distributions goals by leveraging compositionality to imagine new goals from known ones.

This paper presents **I**ntrinsic **M**otivations **A**nd **G**oal **IN**vention for **E**xploration (IMAGINE): a learning architecture which leverages natural language (NL) interactions with a descriptive social partner (SP) to explore procedurally-generated scenes and interact with objects. IMAGINE discovers meaningful environment interactions through its own exploration (Figure 1a) and episode-level NL descriptions provided by SP (1b). These descriptions are turned into targetable goals by the agent (1c). The agent learns to represent goals by jointly training a language encoder mapping NL to goal embeddings and a goal-achievement reward function (1d). The latter evaluates whether the current scene satisfies any given goal. These signals (ticks in Figure 1d-e) are then used as training signals for policy learning. More importantly, IMAGINE can invent new goals by composing known ones (1f). Its internal goal-achievement function allows it to train autonomously on these imagined goals.

**Related work.** The idea that language understanding is grounded in one's experience of the world and should not be secluded from the perceptual and motor systems has a long history in Cognitive Science [30, 74]. This vision was transposed to intelligent systems [66, 50], applied to human-machine interaction [24, 49] and recently to deep RL via frameworks such as *BabyAI* [17].

In their review of *RL algorithms informed by NL*, Luketina et al. [48] distinguish between *language-conditional* problems where language is required to solve the task and *language-assisted* problems

where language is a supplementary help. In the first category, most works propose instruction-following agents [59, 15, 4, 21, 40, 33, 20]. Although our system is *language-conditioned*, it is not *language-instructed*: it is never given any instruction or reward but sets its own goals and learns its own internal reward function. Bahdanau et al. [4] and Fu et al. [28] also learn a reward function but require extensive expert knowledge (expert dataset and known environment dynamics respectively), whereas our agent uses experience generated by its own exploration.

Language is also particularly well suited for Hindsight Experience Replay [2]: descriptions of the current state can be used to relabel trajectories, enabling agents to transfer skills across goals. While previous works used a hard-coded descriptive function [13, 40] or trained a generative model [20] to generate goal substitutes, we leverage the learned reward function to scan goal candidates.

To our knowledge, no previous work has considered the use of compositional goal imagination to enable creative exploration of the environment. The linguistic basis of our goal imagination mechanism is grounded in construction grammar (CG). CG is a usage-based approach that characterizes language acquisition as a trajectory starting with pattern imitation and the discovery of equivalence classes for argument substitution, before evolving towards the recognition and composition of more abstract patterns [68, 31]. This results in a structured inventory of constructions as form-to-meaning mappings that can be combined to create novel utterances [31]. The discovery and substitution of equivalent words in learned schemas is observed directly in studies of child language [70, 68]. Computational implementations of this approach have demonstrated its ability to foster generalization [38] and was also used for data augmentation to improve the performance of neural seq2seq models in NLP [1].

Imagining goals by composing known ones only works in association with *systematic generalization* [5, 37]: generalizations of the type *grow any animal + grasp any plant → grow any plant*. These were found to emerge in instruction-following agents, including generalizations to new combinations of motor predicates, object colors and shapes [36, 37, 4]. Systematic generalization can occur when objects share common attributes (e.g. type, color). We directly encode that assumption into our models by representing objects as *single-slot object files* [34]: separate entities characterized by shared attributes. Because all objects have similar features, we introduce a new object-centered inductive bias: object-based modular architectures based on Deep Sets [73].

**Contributions.** This paper introduces:

1. The concept of imagining new goals using language compositionality to drive exploration.

2. IMAGINE: an intrinsically motivated agent that uses goal imagination to explore its environment, discover and master object interactions by leveraging NL descriptions from a social partner.

3. Modular policy and reward function with systematic generalization properties enabling IMAGINE to train on imagined goals. Modularity is based on Deep Sets, gated attention mechanisms and object-centered representations.

4. *Playground*: a procedurally-generated environment designed to study several types of generalizations (across predicates, attributes, object types and categories).

5. A study of IMAGINE investigating: 1) the effects of our goal imagination mechanism on generalization and exploration; 2) the identification of general properties of imagined goals required for any algorithm to have a similar impact; 3) the impact of modularity and 4) social interactions.

## 2  Problem Definition

**Open-ended learning environment.** We consider a setup where agents evolve in an environment filled with objects and have no prior on the set of possible interactions. An agent decides what and when to learn by setting its own goals, and has no access to external rewards.

However, to allow the agent to learn relevant skills, a social partner (SP) can watch the scene and plays the role of a human caregiver. Following a developmental approach [3], we propose a hard-coded surrogate SP that models important aspects of the developmental processes seen in humans:

- At the beginning of each episode, the agent chooses a goal by formulating a sentence. SP then provides agents with optimal learning opportunities by organizing the scene with: 1)

the required objects to reach the goal (not too difficult) 2) procedurally-generated distracting objects (not too easy and providing further discovery opportunities). This constitutes a developmental scaffolding modelling the process of Zone of Proximal Development (ZPD) introduced by Vygotsky to describe infant-parent learning dynamics [72].

- At the end of each episode, SP utters a set of sentences describing achieved and meaningful outcomes (except sentences from a test set). Linguistic guidance given through descriptions are a key component of how parents "teach" language to infants, which contrasts with instruction following (providing a linguistic command and then a reward), that is rarely seen in real parent-child interactions [69, 9]. By default, SP respects the 3 following properties: *precision*: descriptions are accurate, *exhaustiveness*: it provides all valid descriptions for each episode and *full-presence*: it is always available. Section 4.4 investigates relaxations of the last two assumptions.

Pre-verbal infants are known to acquire object-based representations very early [65, 41] and, later, to benefit from a simplified parent-child language during language acquisition [51]. Pursuing a developmental approach [3], we assume corresponding object-based representations and a simple grammar. As we aim to design agents that bootstrap creative exploration without prior knowledge of possible interactions or language, we do not consider the use of pre-trained language models.

**Evaluation metrics.** This paper investigates how goal imagination can lead agents to efficiently and creatively explore their environment to discover interesting interactions with objects around. In this quest, SP guides agents towards a set of interesting outcomes by uttering NL descriptions. Through compositional recombinations of these sentences, goal imagination aims to drive creative exploration, to push agents to discover outcomes beyond the set of outcomes known by SP. We evaluate this desired behavior by three metrics: 1) the generalization of the policy to new states, using goals from the training set that SP knows and describes; 2) the generalization of the policy to new language goals, using goals from the testing set unknown to SP; 3) goal-oriented exploration metrics. These measures assess the quality of the agents' intrinsically motivated exploration. Measures 1) and 2) are also useful to assess the abilities of agents to learn language skills. We measure generalization for each goal as the success rate over 30 episodes and report $\overline{SR}$ the average over goals. We evaluate exploration with the *interesting interaction count* (I2C). I2C is computed on different sets of interesting interactions: behaviors a human could infer as goal-directed. These sets include the training, testing sets and an extra set containing interactions such as bringing water or food to inanimate objects. $I2C_{\mathcal{I}}$ measures the number of times interactions from $\mathcal{I}$ were observed over the last epoch (600 episodes), whether they were targeted or not (see Supplementary Section 3). Thus, I2C measures the penchant of agents to explore interactions with objects around them. Unless specified otherwise, we provide means $\mu$ and standard deviations over 10 seeds and report statistical significance using a two-tail Welch's t-test with null hypothesis $\mu_1 = \mu_2$, at level $\alpha = 0.05$ (noted by star and circle markers in figures) [23].

## 3 Methods

### 3.1 The *Playground* environment

We argue that the study of new mechanisms requires the use of controlled environments. We thus introduce *Playground*, a simple environment designed to study the impact of goal imagination on exploration and generalization by disentangling it from the problems of perception and fully-blown NL understanding. The *Playground* environment is a continuous 2D world, with procedurally-generated scenes containing $N = 3$ objects, from 32 different object types (*e.g. dog, cactus, sofa, water, etc.*), organized into 5 categories (*animals, furniture, plants, etc*), see Figure 1. To our knowledge, it is the first environment that introduces object categories and category-dependent combinatorial dynamics, which allows the study of new types of generalization. We release *Playground* in a separate repository.[2]

**Agent perception and embodiment.** Agents have access to state vectors describing the scene: the agent's body and the objects. Each object is represented by a set of features describing its type, position, color, size and whether it is grasped. Categories are not explicitly encoded. Objects are made unique by the procedural generation of their color and size. The agent can perform bounded

translations in the 2D plane, grasp and release objects with its gripper. It can make animals and plants grow by bringing them the right supply (food or water for animals, water for plants).

**Grammar.** The following grammar generates the descriptions of the 256 achievable goals ($\mathcal{G}^{\text{A}}$):

1. Go: *<go + **zone**>* *(e.g. go bottom left)*
2. Grasp: *< grasp + any + **color** + thing>* *(e.g. grasp any blue thing)* OR
   *<grasp + **color** ∪ {any} + **object type** ∪ **object category**>* *(e.g. grasp red cat)*
3. Grow: *<grow + any + **color** + thing>* *(e.g. grow any red thing)* OR
   *<grow + **color** ∪ {any} + **living thing** ∪ {living_thing, animal, plant}>* *(e.g. grow green animal)*

**Bold** and { } are sets of words while *italics* are specific words. The grammar is structured around the 3 predicates *go*, *grasp* and *grow*. Objects can be referred to by a combination of their color and either their object name or category, or simply by one of these. The set of achievable goals is partitioned into *training* ($\mathcal{G}^{\text{train}}$) and *testing* ($\mathcal{G}^{\text{test}}$). $\mathcal{G}^{\text{test}}$ maximizes the compound divergence with a null atom divergence with respect to $\mathcal{G}^{\text{train}}$: testing sentences (compounds) are out of the distribution of $\mathcal{G}^{\text{train}}$ sentences, but their words (atoms) belong to the distribution of words in $\mathcal{G}^{\text{train}}$ [44]. SP only provides descriptions from $\mathcal{G}^{\text{train}}$. We limit the set of goals to better control the complexity of our environment and enable a careful study of the generalization properties. Supplementary Section 1 provides more details about the environment, the grammar and SP as well as the pseudo-code of our learning architecture.

## 3.2 The IMAGINE Architecture

IMAGINE agents build a repertoire of goals and train two internal models: 1) a goal-achievement reward function $\mathcal{R}$ to predict whether a given description matches a behavioral trajectory; 2) a policy $\pi$ to achieve behavioral trajectories matching descriptions. The architecture is presented in Figure 2 and follows this logic:

1. The *Goal Generator* samples a target goal $g_{\text{target}}$ from known and imagined goals ($\mathcal{G}_{\text{known}} \cup \mathcal{G}_{\text{im}}$).
2. The agent (*RL Agent*) interacts with the environment using its policy $\pi$ conditioned on $g_{\text{target}}$.
3. State-action trajectories are stored in a replay buffer $mem(\pi)$.
4. SP's descriptions of the last state are considered as potential goals $\mathcal{G}_{\text{SP}}(\mathbf{s}_T) = \mathcal{D}_{\text{SP}}(\mathbf{s}_T)$.
5. $mem(\mathcal{R})$ stores positive pairs $(\mathbf{s}_T, \mathcal{G}_{\text{SP}}(\mathbf{s}_T))$ and infers negative pairs $(\mathbf{s}_T, \mathcal{G}_{\text{known}} \setminus \mathcal{G}_{\text{SP}}(\mathbf{s}_T))$.
6. The agent then updates:
   - *Goal Gen.*: $\mathcal{G}_{\text{known}} \leftarrow \mathcal{G}_{\text{known}} \cup \mathcal{G}_{\text{SP}}(\mathbf{s}_T)$ and $\mathcal{G}_{im} \leftarrow \text{Imagination}(\mathcal{G}_{\text{known}})$.
   - *Language Encoder* ($L_e$) and *Reward Function* ($\mathcal{R}$) are updated using data from $mem(\mathcal{R})$.
   - *RL agent*: We sample a batch of state-action transitions $(\mathbf{s}, \mathbf{a}, \mathbf{s}')$ from $mem(\pi)$. Then, we use *Hindsight Replay* and $\mathcal{R}$ to bias the selection of substitute goals to train on ($g_s$) and compute the associated rewards $(\mathbf{s}, \mathbf{a}, \mathbf{s}', g_s, r)$. Substituted goals $g_s$ can be known or imagined goals. Finally, the policy and critic are trained via RL.

**Goal generator.** It is a generative model of NL goals. It generates target goals $g_{\text{target}}$ for data collection and substitutes goals $g_s$ for hindsight replay. When goal imagination is disabled, the goal generator samples uniformly from the set of known goals $\mathcal{G}_{\text{known}}$, sampling random vectors if empty. When enabled, it samples with equal probability from $\mathcal{G}_{\text{known}}$ and $\mathcal{G}_{\text{im}}$ (set of imagined goals). $\mathcal{G}_{\text{im}}$ is generated using a mechanism grounded in construction grammar that leverages the compositionality of language to imagine new goals from $\mathcal{G}_{\text{known}}$. The heuristic consists in computing sets of *equivalent words*: words that appear in two sentences that only differ by one word. For example, from *grasp red lion* and *grow red lion*, *grasp* and *grow* can be considered *equivalent* and from *grasp green tree* one can imagine a new goal *grow green tree* (see Figure 1f). Imagined goals do not include known goals. Among them, some are meaningless, some are syntactically correct but infeasible (e.g. *grow red lamp*) and some belong to $\mathcal{G}^{\text{test}}$, or even to $\mathcal{G}^{\text{train}}$ before they are encountered by the agent and described by SP. The pseudo-code and all imaginable goals are provided in Supplementary Section 4.

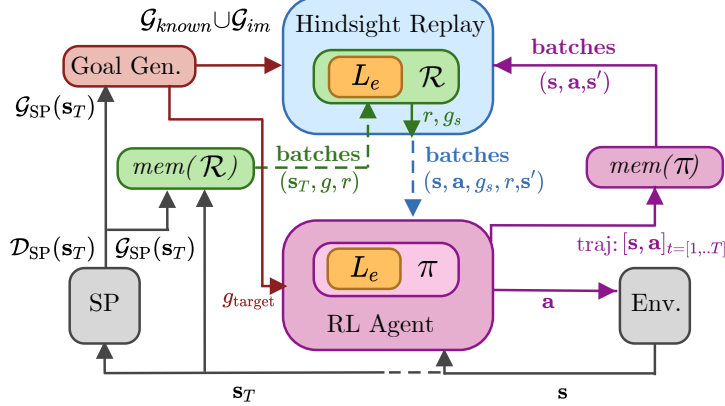

Figure 2: **IMAGINE architecture.** Colored boxes show the different modules of IMAGINE. Lines represent update signals (dashed) and function outputs (plain). The language encoder $L_e$ is shared.

**Language encoder.** The language encoder ($L_e$) embeds NL goals ($L_e : \mathcal{G}^{\text{NL}} \rightarrow \mathbb{R}^{100}$) using an LSTM [39] trained jointly with the reward function. $L_e$ acts as a goal translator, turning the goal-achievement reward function, policy and critic into language-conditioned functions.

**Object-centered modular architectures.** The goal-achievement reward function, policy and critic leverage novel *modular-attention* (MA) architectures based on Deep Sets [73], gated attention mechanisms [14] and object-centered representations. The idea is to ensure efficient skill transfer between objects, no matter their position in the state vector. This is done through the combined use of a shared neural network that encodes object-specific features and a permutation-invariant function to aggregate the resulting latent encodings. The shared network independently encodes, for each object, an affordance between this object (object observations), the agent (body observations) and its current goal. The goal embedding, generated by $L_e$, is first cast into an attention vector in $[0, 1]$, then fused with the concatenation of object and body features via an Hadamard product (gated-attention [14]). The resulting object-specific encodings are aggregated by a permutation-invariant function and mapped to the desired output via a final network (e.g. into actions or action-values). Supplementary Section 5 provides visual representations.

**Reward function.** Learning a goal-achievement reward function ($\mathcal{R}$) is framed as binary classification: $\mathcal{R}(\mathbf{s}, \mathbf{g}) : \mathcal{S} \times \mathbb{R}^{100} \rightarrow \{0, 1\}$. We use the MA architecture with attention vectors $\boldsymbol{\alpha}^g$, a shared network $\text{NN}^{\mathcal{R}}$ with output size 1 and a logical OR aggregation. $\text{NN}^{\mathcal{R}}$ computes object-dependent rewards $r_i$ in $[0, 1]$ from the object-specific inputs and the goal embedding. The final binary reward is computed by $\text{NN}^{\text{OR}}$ which outputs 1 whenever $\exists j : r_j > 0.5$. We pre-trained a neural-network-based OR function to enable end-to-end training with back-propagation. The overall function is:

$$\mathcal{R}(\mathbf{s}, g) = \text{NN}^{\text{OR}}([\text{NN}^{\mathcal{R}}(\mathbf{s}_{obj(i)} \odot \boldsymbol{\alpha}^g)]_{i \in [1..N]})$$

*Data.* Interacting with the environment and SP, the agent builds a set of entries $[\mathbf{s}_T, \ g, \ r]$ with $g \in \mathcal{G}_{\text{known}}$ where $r \in \{0, 1\}$ rewards the achievement of $g$ in state $\mathbf{s}_T$: $r = 1$ if $g \in \mathcal{G}_{\text{SP}}(\mathbf{s}_T)$ and 0 otherwise. $L_e$ and $\mathcal{R}$ are periodically updated jointly by back-propagation on this dataset.

**Multi-goal RL agent.** Our agent is controlled by a goal-conditioned policy $\pi$ [62] based on the MA architecture (see Supplementary Figure 9b). It uses an attention vector $\boldsymbol{\beta}^g$, a shared network $\text{NN}^\pi$, a sum aggregation and a mapper $\text{NN}^a$ that outputs the actions. Similarly, the critic produces action-values via $\boldsymbol{\gamma}^g$, $\text{NN}^Q$ and $\text{NN}^{\text{a-v}}$ respectively:

$$\pi(\mathbf{s}, g) = \text{NN}^a \Big( \sum_{i \in [1..N]} \text{NN}^\pi(\mathbf{s}_{obj(i)} \odot \boldsymbol{\beta}^g) \Big) \qquad Q(\mathbf{s}, \mathbf{a}, g) = \text{NN}^{\text{a-v}} \Big( \sum_{i \in [1..N]} \text{NN}^Q([\mathbf{s}_{obj(i)}, \mathbf{a}] \odot \boldsymbol{\gamma}^g) \Big).$$

Both are trained using DDPG [47], although any other off-policy algorithm can be used. As detailed in Supplementary Section 6, our agent uses a form of Hindsight Experience Replay [2].

# 4 Experiments and Results

This section first showcases the impact of goal imagination on exploration and generalization (Section 4.1). For a more complete picture, we analyze other goal imagination mechanisms and investigate the properties enabling these effects (Section 4.2). Finally, we show that our modular architectures are crucial to a successful goal imagination (Section 4.3) and discuss more realistic interactions with SP (Section 4.4). IMAGINE agents achieve near perfect generalizations to new states (training set of goals): $\overline{\text{SR}} = 0.95 \pm 0.05$. We thus focus on language generalization and exploration. Supplementary Sections 2 to 7 provide additional results and insights organized by theme (Generalization, Exploration, Goal Imagination, Architectures, Reward Function and Visualizations).

## 4.1 How does Goal Imagination Impact Generalization and Exploration?

**Global generalization performance.** Figure 3a shows $\overline{\text{SR}}$ on the set of testing goals, when the agent starts imagining new goals early (after $6 \cdot 10^3$ episodes), half-way (after $48 \cdot 10^3$ episodes) or when not allowed to do so. Imagining goals leads to significant improvements in generalization.

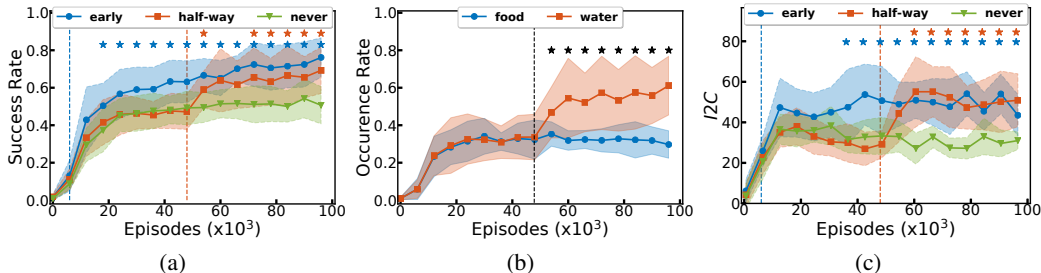

Figure 3: **Goal imagination drives exploration and generalization.** Vertical dashed lines mark the onset of goal imagination. (a) $\overline{\text{SR}}$ on testing set. (b) Behavioral adaptation, empirical probabilities that the agent brings supplies to a plant when trying to grow it. (c) I2C computed on the testing set. Stars indicate significance (a and c are tested against *never*).

**A particular generalization: growing plants.** Agents learn to grow animals from SP's descriptions, but are never told they could grow plants. When evaluated offline on the *growing-plants* goals before goal imagination, agents' policies perform a sensible zero-shot generalization and bring them water or food with equal probability, as they would do for animals (Figure 3b, left). As they start to imagine and target these goals, their behavior adapts (Figure 3b, right). If the reward function shows good zero-shot abilities, it only provides positive rewards when the agent brings water. The policy slowly adapts to this internal reward signal and pushes agents to bring more water. We call this phenomenon *behavioral adaptation*. Supplementary Section 2 details the generalization abilities of IMAGINE for 5 different types of generalizations involving predicates, attributes and categories.

**Exploration.** Figure 3c presents the I2C metric computed on the set of interactions related to $\mathcal{G}^{\text{test}}$ and demonstrates the exploration boost triggered by goal imagination. Supplementary Section 3 presents other I2C metrics computed on additional interactions sets.

## 4.2 What If We Used Other Goal Imagination Mechanisms?

**Properties of imagined goals.** We propose to characterize goal imagination mechanisms by two properties: 1) *Coverage*: the fraction of $\mathcal{G}^{\text{test}}$ found in $\mathcal{G}_{\text{im}}$ and 2) *Precision*: the fraction of the imagined goals that are achievable. We compare our goal imagination mechanism based on the construction grammar heuristic (CGH) to variants characterized by 1) lower coverage; 2) lower precision; 3) perfect coverage and precision (oracle); 4) random goal imagination baseline (random sequences of words from $\mathcal{G}^{\text{train}}$ leading to near null coverage and precision). These measures are computed at the end of experiments, when all goals from $\mathcal{G}^{\text{train}}$ have been discovered (Figure 4a).

Figure 4b shows that CGH achieves a generalization performance on par with the oracle. Reducing the coverage of the goal imagination mechanism still brings significant improvements in generalization. Supplementary Section 4 shows, for the *Low Coverage* condition, that the generalization performance

on the testing goals that were imagined is not statistically different from the performance on similar testing goals that could have been imagined but were not. This implies that the generalization for imagined goals also benefits similar non-imagined goals from $\mathcal{G}^{\text{test}}$. Finally, reducing the precision of imagined goals (gray curve) seems to impede generalization (no significant difference with the *no imagination* baseline). Figure 4c shows that all goal imagination heuristics enable a significant exploration boost. The random goal baseline acts as a control condition. It demonstrates that the generalization boost is not due to a mere effect of network regularization introduced by adding random goals (no significant effect w.r.t. the *no imagination* baseline). In the same spirit, we also ran a control using random goal embeddings, which did not produce any significant effects.

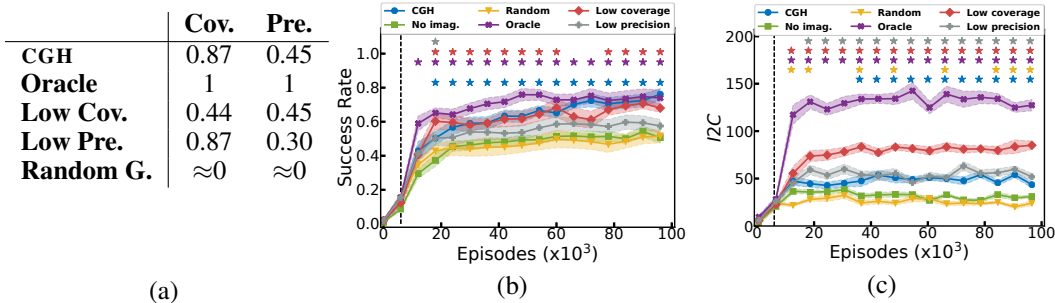

|  | Cov. | Pre. |
|---|---|---|
| **CGH** | 0.87 | 0.45 |
| **Oracle** | 1 | 1 |
| **Low Cov.** | 0.44 | 0.45 |
| **Low Pre.** | 0.87 | 0.30 |
| **Random G.** | $\approx 0$ | $\approx 0$ |

(a)           (b)           (c)

Figure 4: **Goal imagination properties.** (a) Coverage and precision of different goal imagination heuristics. (b) $\overline{\text{SR}}$ on testing set. (c) I2C on $\mathcal{G}^{\text{test}}$. We report *sem* (standard error of the mean) instead of *std* to improve readability. Stars indicate significant differences w.r.t the *no imagination* condition.

## 4.3 How Does Modularity Interact with Goal Imagination?

We compared MA to flat architectures (FA) that consider the whole scene at once. As the use of FA for the reward function showed poor performance on $\mathcal{G}^{\text{train}}$, Table 1 only compares the use of MA and FA for the policy. MA shows stronger generalization and is the only architecture allowing an additional boost with goal imagination. Only MA policy architectures can leverage the novel reward signals coming from imagined goals and turn them into *behavioral adaptation*. Supplementary Section 5 provides additional details.

Table 1: Policy architectures performance. $\overline{\text{SR}}_{\text{test}}$ at convergence.

|  | MA* | FA |
|---|---|---|
| Im. | $0.76 \pm 0.1$ | $0.15 \pm 0.05$ |
| No Im. | $0.51 \pm 0.1$ | $0.17 \pm 0.04$ |
| p-val | 4.8e-5 | 0.66 |

## 4.4 Can We Use More Realistic Feedbacks?

We study the relaxation of the *full-presence* and *exhaustiveness* assumptions of SP. We first relax *full-presence* while keeping *exhaustiveness* (blue, yellow and purple curves). When SP has a 10% chance of being present (yellow), imaginative agents show generalization performance on par with the unimaginative agents trained in a full-presence setting (green), see Figure 5. However, when the same amount of feedback is concentrated in the first 10% episodes (purple), goal imagination enables significant improvements in generalization (w.r.t. green). This is reminiscent of children who require less and less attention as they grow into adulthood and is consistent with Chan et al. [13]. Relaxing *exhaustiveness*, SP only provides one positive and one negative description every episode (red) or in 50% of the episodes (gray). Then, generalization performance matches the one of unimaginative agents in the exhaustive setting (green).

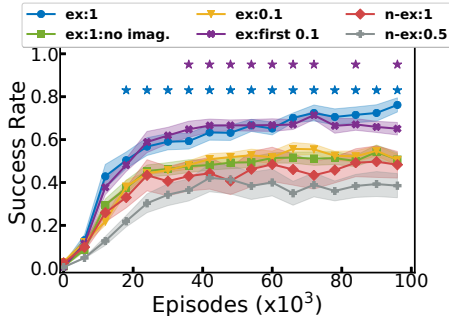

Figure 5: **Influence of social feedbacks.** $\overline{\text{SR}}$ on $\mathcal{G}^{\text{test}}$ for different social strategies. Stars indicate significant differences w.r.t. *ex:1 no imag..* sem plotted, 5 seeds.

# 5 Discussion and Conclusion

IMAGINE is a learning architecture that enables autonomous learning by leveraging NL interactions with a social partner. As other algorithms from the IMGEP family, IMAGINE sets its own goals and builds behavioral repertoires without external rewards. As such, it is distinct from traditional instruction-following RL agents. This is done through the joint training of a language encoder for goal representation and a goal-achievement reward function to generate internal rewards. Our proposed modular architectures with gated-attention enable efficient out-of-distribution generalization of the reward function and policy. The ability to imagine new goals by composing known ones leads to further improvements over initial generalization abilities and fosters exploration beyond the set of interactions relevant to SP. Our agent even tries to grow pieces of furniture with supplies, a behavior that can echo the way a child may try to feed his doll.

IMAGINE does not need externally-provided rewards but learns which behaviors are *interesting* from language-based interactions with SP. In contrast with hand-crafted reward functions, NL descriptions provide an easy way to guide machines towards relevant interactions. *A posteriori* counterfactual feedback is easier to communicate for humans, especially when possible effects are unknown and, thus, the set of possible instructions is undefined. Hindsight learning also greatly benefits from such counterfactual feedback and improves sample efficiency. Attention mechanisms further extend the interpretability of the agent's learning by mapping language to attentional scaling factors (see Supplementary Figure 12). In addition, Section 4.4 shows that agents can learn to achieve goals from a relatively small number of descriptions, paving the way towards human-provided descriptions.

*Playground* is a tool that we hope will enable the community to further study under-explored descriptive setups with rich combinatorial dynamics, as well as goal imagination. It is designed for the study of goal imagination and combinatorial generalization. Compared to existing environments [36, 17, 13], we allow the use of descriptive feedback, introduce the notion of object categories and category-dependent object interactions (*Grow* refer to different modalities for *plants* or *animals*). Playground can easily be extended by adding objects, attributes, category- or object-type-dependent dynamics.

IMAGINE could be combined with unsupervised multi-object representation learning algorithms [11, 35] to work directly from pixels, practically enforcing object-centered representations. The resulting algorithm would still be different from goal-as-state approaches [53, 58, 52]. Supplementary Section 8 discusses the relevance of comparing IMAGINE to these works. Some tasks involve instruction-based navigation in visual environments that do not explictly represent objects [54, 64]. Here, also, imagining new instructions from known ones could improve exploration and generalization. Finally, we believe IMAGINE could provide interesting extensions in hierarchical settings, like in Jiang et al. [40], with novel goal imagination boosting low-level exploration.

**Future work.** A more complex language could be introduced, for example, by considering object relationships (e.g. *Grasp any X left of Y*), see [43] for a preliminary experiment in this direction. While the use of pre-trained language models [61] does not follow our developmental approach, it would be interesting to study how they would interact with goal imagination. Because CGH performs well in our setup with a medium precision (0.45) and because similar mechanisms were successfully used for data augmentation in complex NLP tasks [1], we believe our goal imagination heuristic could scale to more realistic language.

We could reduce the burden on SP by considering unreliable feedbacks (lower precision), or by conditioning goal generation on the initial scene (e.g. using mechanisms from Cideron et al. [20]). One could also add new interaction modalities by letting SP make demonstrations, propose goals or guide the agent's attention. Our modular architectures, because they are set functions, could also directly be used to consider variable numbers of objects. Finally, we could use off-policy learning [29] to reinterpret past experience in the light of new imagined goals without any additional environment interactions.

**Links.** Demonstration videos are available at https://sites.google.com/view/imagine-drl. The source code of playground environment can be found at https://github.com/flowersteam/playground_env and the source code of the IMAGINE architecture https://github.com/flowersteam/Imagine.

## Broader Impact Statement

We present a reinforcement learning architecture where autonomous agents interact with a social partner to explore a large set of possible interactions and learn to master them. As a result, our work contributes to facilitating human intervention in the learning process of a robot, which we believe is a key step towards more explainable and safer autonomous robots. Besides, by releasing our code, we believe that we help efforts in reproducible science and allow the wider community to build upon and extend our work in the future. In that spirit, we also provide clear explanations on the number of seeds, error bars, and statistical testing when reporting the results.

## Acknowledgments and Disclosure of Funding

Cédric Colas and Tristan Karch are partly funded by the French Ministère des Armées - Direction Générale de l'Armement. Nicolas Lair is supported by ANRT/CIFRE contract No. 151575A20 from Cloud Temple.

## Footnotes

[2] https://github.com/flowersteam/playground_env

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
