[Supplementary Material]

## Supplementary Material

This supplementary material provides additional methods, results and discussion, as well as implementation details.

- Section 1 gives a complete description of our setup and of the *Playground* environment.
- Section 2 presents a focus on generalization and studies different types of generalization.
- Section 3 presents a focus on exploration and how it is influenced by goal imagination.
- Section 4 presents a focus on the goal imagination mechanism we use for IMAGINE.
- Section 5 presents a focus on the *Modular-Attention* architecture.
- Section 6 presents a focus on the benefits of learning the reward function.
- Section 7 provides additional visualization of the goal embeddings and the attention vectors.
- Section 8 discusses the comparison with goal-as-state approaches.
- Section 9 gives all necessary implementation details.

## 1 Complete Description of the Playground Environment and Its Language

**Environment description.** The environment is a 2D square: $[-1.2, 1.2]^2$. The agent is a disc of diameter $0.05$ with an initial position $(0,0)$. Objects have sizes uniformly sampled from $[0.2, 0.3]$ and their initial positions are randomized so that they are not in contact with each other. The agent has an action space of size 3 bounded in $[-1, 1]$. The first two actions control the agent's continuous 2D translation (bounded to $0.15$ in any direction). The agent can grasp objects by getting in contact with them and closing its gripper (positive third action), unless it already has an object in hand. Objects include 10 animals, 10 plants, 10 pieces of furniture and 2 supplies. Admissible categories are *animal, plant, furniture, supply* and *living_thing* (animal or plant), see Figure 1. Objects are assigned a color attribute (red, blue or green). Their precise color is a continuous RGB code uniformly sampled from RGB subspaces associated with their attribute color. Each scene contains 3 of these procedurally-generated objects (see paragraph about the Social Partner below).

Figure 1: Representation of possible objects types and categories.

**Agent perception.** At time step $t$, we can define an observation $\mathbf{o}_t$ as the concatenation of body observations (2D-position, gripper state) and objects' features. These two types of features form affordances between the agent and the objects around. These affordances are necessary to understand the meaning of object interactions like *grasp*. The state $\mathbf{s}_t$ used as input of the models is the concatenation of $\mathbf{o}_t$ and $\Delta\mathbf{o}_t = \mathbf{o}_t - \mathbf{o}_0$ to provide a sense of time. This is required to acquire the understanding and behavior related to the *grow* predicate, as the agent needs to observe and produce a change in the object's size.

**Social Partner.** SP has two roles:

- *Scene organization*: SP organize the scene according to the goal selected by the agent. When the agent selects a goal, it communicates it to SP. If the goal starts by the word *grow*, SP adds a procedurally-generated supply (water or food for animals, water for plants) of any size and color to the scene. If the goal contains an object (e.g. *red cat*), SP adds a corresponding object to the scene (with a procedurally generated size and RGB color). Remaining objects are generated procedurally. As a result, the objects required to fulfill a goal are always

present and the scene contains between 1 (*grow* goals) and 3 (*go* goals) random objects. Note that all objects are procedurally generated (random initial position, RGB color and size).

- *Scene description*: SP provides NL descriptions of interesting outcomes experienced by the agent at the end of episodes. It takes the final state of an episode ($\mathbf{s}_T$) as input and returns matching NL descriptions: $\mathcal{D}_{\text{SP}}(\mathbf{s}_T) \subset \mathcal{D}^{\text{SP}}$. When SP provides *descriptions*, the agent considers them as targetable *goals*. This mapping $\mathcal{D}^{\text{SP}} \rightarrow \mathcal{G}^{\text{train}}$ simply consists in removing the first *you* token (e.g. turning *you grasp red door* into the goal *grasp red door*). Given the set of previously discovered goals ($\mathcal{G}_{\text{known}}$) and new descriptions $\mathcal{D}_{\text{SP}}(\mathbf{s}_T)$, the agent infers the set of goals that were not achieved: $\mathcal{G}_{\text{na}}(\mathbf{s}_T) = \mathcal{G}_{\text{known}} \setminus \mathcal{D}_{\text{SP}}(\mathbf{s}_T)$, where $\setminus$ indicates the complement.

**Grammar.** We now present the grammar that generates descriptions for the set of goals achievable in the Playground environment ($\mathcal{G}^A$). **Bold** and { } refer to sets of words while *italics* refers to particular words:

1. Go: *(e.g. go bottom left)*
   - *go* + **zone**
2. Grasp: *(e.g. grasp any animal)*
   - *grasp* + **color** ∪ {*any*} + **object type** ∪ **object category**
   - *grasp* + *any* + **color** + *thing*
3. Grow: *(e.g. grow blue lion)*
   - *grow* + **color** ∪ {*any*} + **living thing** ∪ {*living_thing, animal, plant*}
   - *grow* + *any* + **color** + *thing*

Word sets are defined by:

- **zone** = {*center, top, bottom, right, left, top left, top right, bottom left, bottom right*}
- **color** = {*red, blue, green*}
- **object type** = **living thing** ∪ **furniture** ∪ **supply**
- **object category** = {*living_thing, animal, plant, furniture, supply*}
- **living thing** = **animal** ∪ **plant**
- **animal** = {*dog, cat, chameleon, human, fly, parrot, mouse, lion, pig, cow*}
- **plant** = {*cactus, carnivorous, flower, tree, bush, grass, algae, tea, rose, bonsai*}
- **furniture** = {*door, chair, desk, lamp, table, cupboard, sink, window, sofa, carpet*}
- **supply** = {*water, food*}
- **predicate** = {*go, grasp, grow*}

We partition this set of achievable goals into a training ($\mathcal{G}^{\text{train}}$) and a testing ($\mathcal{G}^{\text{test}}$) set. Goals from $\mathcal{G}^{\text{test}}$ are intended to evaluate the ability of our agent to explore the set of achievable outcomes beyond the set of outcomes described by SP. The next section introduces this testing set and focuses on generalization. Note that some goals might be syntactically valid but not achievable. This includes all goals of the form *grow* + **color** ∪ {*any*} + **furniture** ∪ {*furniture*} (e.g. *grow red lamp*).

**IMAGINE Pseudo-Code.** Algorithm 1 outlines the pseudo-code of our learning architecture. See Main Section 3.2 for high-level descriptions of each module and function.

**Algorithm 1:** IMAGINE

---

1: **Input:** env, SP
2: **Initialize:** $L_e$, $\mathcal{R}$, $\pi$, $mem(\mathcal{R})$, $mem(\pi)$, $\mathcal{G}_{\text{known}}$, $\mathcal{G}_{\text{im}}$
            # Random initializations for networks
            # empty sets for memories and goal sets
3: **for** $e = 1 : N_{episodes}$ **do**
4:     **if** $\mathcal{G}_{\text{known}} \neq \emptyset$ **then**
5:         sample $g_{\text{NL}}$ from $\mathcal{G}_{\text{known}} \cup \mathcal{G}_{\text{im}}$
6:         $g \leftarrow L_e(g_{\text{NL}})$
7:     **else**
8:         sample $g$ from $\mathcal{N}(0, \mathbf{I})$
9:     $s_0 \leftarrow$ env.reset()
10:     **for** $t = 1 : T$ **do**
11:         $a_t \leftarrow \pi(s_{t-1}, g)$
12:         $s_t \leftarrow$ env.step($a_t$)
13:         $mem_\pi$.add($s_{t-1}, a_t, s_t$)
14:     $\mathcal{G}_{\text{SP}} \leftarrow$ SP.get_descriptions($s_T$)
15:     $\mathcal{G}_{\text{known}} \leftarrow \mathcal{G}_{\text{known}} \cup \mathcal{G}_{\text{SP}}$
16:     $mem(\mathcal{R})$.add($s_T, g_{\text{NL}}$) for $g_{\text{NL}}$ in $\mathcal{G}_{\text{SP}}$
17:     **if** goal imagination allowed **then**
18:         $\mathcal{G}_{\text{im}} \leftarrow$ **Imagination**($\mathcal{G}_{\text{known}}$) # see Algorithm 2
19:     Batch$_\pi \leftarrow$ **ModularBatchGenerator**($mem(\pi)$)      # Batch$_\pi = \{(s, a, s')\}$
20:     Batch$_\pi \leftarrow$ **Hindsight**(Batch$_\pi, \mathcal{R}, \mathcal{G}_{\text{known}}, \mathcal{G}_{\text{im}}$)   # Batch$_\pi = \{(s, a, r, g, s')\}$ where $r = \mathcal{R}(s, g)$
21:     $\pi \leftarrow$ **RL_Update**(Batch$_\pi$)
22:     **if** $e \% \text{reward\_update\_freq} == 0$ **then**
23:         Batch$_\mathcal{R} \leftarrow$ ModularBatchGenerator($mem(\mathcal{R})$)
24:         $L_e, \mathcal{R} \leftarrow$ **LE&RewardFunctionUpdate**(Batch$_\mathcal{R}$)

---

## 2 Focus on Generalization

Because scenes are procedurally-generated, $\overline{\text{SR}}$ computed on $\mathcal{G}^{\text{train}}$ measures the generalization to new states. When computed on $\mathcal{G}^{\text{test}}$, however, $\overline{\text{SR}}$ measures both this state generalization and the generalization to new goal descriptions from $\mathcal{G}^{\text{test}}$. As $\overline{\text{SR}}_{\text{train}}$ is almost perfect, this section focuses solely on generalization in the language space: $\overline{\text{SR}}_{\text{test}}$.

**Different types of generalization.** Generalization can occur in two different modules of the IMAGINE architecture: in the reward function and in the policy. Agents can only benefit from goal imagination when their reward function is able to generalize the meanings of imagined goals from the meanings of known ones. When they do, they can further train on imagined goals, which might, in turn, reinforce the generalization of the policy. This section characterizes different types of generalizations that the reward and policy can both demonstrate.

- Type 1 - *Attribute-object generalization*: This is the ability to accurately associate an attribute and an object that were never seen together before. To interpret the goal *grasp red tree* requires to isolate the *red* and *tree* concepts from other sentences and to combine them to recognize a *red tree*. To measure this ability, we removed from the training set all goals containing the following attribute-object combinations: *{blue door, red tree, green dog}* and added them to the testing set (4 goals).

- Type 2 - *Object identification*: This is the ability to identify a new object from its attribute. We left out of the training set all goals containing the word *flower* (4 goals). To interpret the goal *grasp red flower* requires to isolate the concept of *red* and to transpose it to the unknown object *flower*. Note that in the case of *grasp any flower*, the agent cannot rely on the attribute, and must perform some kind of complement reasoning:"if these are known objects, and that is unknown, then if must be a *flower*".

- Type 3 - *Predicate-category generalization*: This is the ability to interpret a predicate for a category when they were never seen together before. As explained in Section 1, a category regroups a set of objects and is not encoded in the object state vector. It is only a linguistic concept. We left out all goals with the *grasp* predicate and the *animal* category (4 goals). To correctly interpret *grasp any animal* requires to identify objects that belong to the animal category (acquired from "growing *animal*" and "growing animal objects" goals), to isolate the concept of *grasping* (acquired from grasping non-*animal* objects) and to combine the two.

- Type 4 - *Predicate-object generalization*: This is the ability to interpret a predicate for an object when they were never seen together before. We leave out all goals with the *grasp* predicate and the *fly* object (4 goals). To correctly interpret *grasp any fly*, the agent should leverage its knowledge about the *grasp* predicate (acquired from the "grasping non-fly objects" goals) and the *fly* object (acquired from the "growing flies" goals).

- Type 5 - *Predicate dynamics generalization*: This is the ability to generalize the behavior associated with a predicate to another category of objects, for which the dynamics is changed. In the *Playground* environment, the dynamics of *grow* with **animals** and **plants** is a a bit different. **animals** can be grown with *food* and *water* whereas **plants** only grow with *water*. We want to see if IMAGINE can learn the dynamics of *grow* on **animals** and generalize it to **plants**. We left out all goals with the *grow* predicate and any of the **plant** objects, *plant* and *living thing* categories (48 goals). To interpret, *grow any plant*, the agent should be able to identify the **plant** objects (acquired from the "grasping plants" goals) and that objects need supplies (food or water) to *grow* (acquired from the "growing animals" goals). Type 5 is more complex than Type 4 for two reasons: 1) because the dynamics change and 2) because it mixes objects and categories. Note that, by definition, the zero-shot generalization is tested without additional reward signals (before imagination). As a result, even the best zero-shot generalization possible cannot adapt the *grow* behavior from animals to plant and would bring food and water with equal probability $p = 0.5$ for each.

Table 1 provides the exhaustive list of goals used to test each type of generalization.

**Different ways to generalize.** Agent can generalize to out-of-distribution goals (from any of the 5 categories above) in three different ways:

1. *Policy zero-shot generalization*: The policy can achieve the new goal without any supplementary training.

2. *Reward zero-shot generalization*: The reward can tell whether the goal is achieved or not without any supplementary training.

3. *Policy n-shot generalization or behavioral adaptation*: When allowed to imagine goals, IMAGINE agents can use the zero-shot generalization of their reward function to autonomously train their policy to improve on imagined goals. After such training, the policy might show improved generalization performance compared to its zero-shot abilities. We call this performance *n-shot generalization*. The policy received supplementary training, but did not leverage any external supervision, only the zero-shot generalization of its internal reward function. This is crucial to achieve Type 5 generalization. As we said, zero-shot generalization cannot figure out that plants only grow with water. Fine-tuning the policy based on experience and internal rewards enables agents to perform *behavioral adaptation*: adapting their behavior with respect to imagined goals in an autonomous manner (see Main Figure 3b).

Table 1: Testing goals in $\mathcal{G}^{\text{test}}$, by type.

| Type 1 | *Grasp blue door, Grasp green dog, Grasp red tree, Grow green dog* |
|---|---|
| Type 2 | *Grasp any flower, Grasp blue flower, Grasp green flower, Grasp red flower, Grow any flower, Grow blue flower, Grow green flower, Grow red flower* |
| Type 3 | *Grasp any animal, Grasp blue animal, Grasp green animal, Grasp red animal* |
| Type 4 | *Grasp any fly, Grasp blue fly, Grasp green fly, Grasp red fly* |
| Type 5 | *Grow any algae, Grow any bonsai, Grow any bush, Grow any cactus Grow any carnivorous, Grow any grass, Grow any living_thing, Grow any plant Grow any rose, Grow any tea, Grow any tree, Grow blue algae Grow blue bonsai, Grow blue bush, Grow blue cactus, Grow blue carnivorous Grow blue grass, Grow blue living_thing, Grow blue plant, Grow blue rose Grow blue tea, Grow blue tree, Grow green algae, Grow green bonsai Grow green bush, Grow green cactus, Grow green carnivorous, Grow green grass Grow green living_thing, Grow green plant, Grow green rose, Grow green tea Grow green tree, Grow red algae, Grow red bonsai, Grow red bush Grow red cactus, Grow red carnivorous, Grow red grass, Grow red living_thing Grow red plant, Grow red rose, Grow red tea, Grow red tree* |

**Experiments.** Figure 2 presents training and generalization performance of the reward function and policy. We evaluate the generalization of the reward function via its average $F_1$ score on $\mathcal{G}^{\text{test}}$, the generalization of the policy by $\overline{\text{SR}}_{\text{test}}$.

*Reward function zero-shot generalization.* When the reward function is trained in parallel of the policy, we monitor its zero-shot generalization capabilities by computing the $F_1$-score over a dataset collected separately with a trained policy run on goals from $\mathcal{G}^{\text{test}}$ (kept fixed across runs for fair comparisons). As shown in Figure 2a, the reward function exhibits good zero-shot generalization properties over 4 types of generalization after $25 \times 10^3$ episodes. Note that, because we test on data collected with a different RL policy, the $F_1$-scores presented in Figure 2a may not faithfully describe the true generalization of the reward function during co-training.

*Policy zero-shot generalization.* The zero-shot performance of the policy is evaluated in Figure 2b (*no imagination* condition) and in the period preceding goal imagination in Figure 2c and 2d (before vertical dashed line). The policy shows excellent zero-shot generalization properties for Type 1, 3 and 4, average zero-shot generalization on Type 5 and fails to generalize on Type 2. Type 1, 3 and 4 can be said to have similar levels of difficulty, as they all require to learn two concepts individually before combining them at test time. Type 2 is much more difficult as the meaning of only one word is known. The language encoder indeed receives a new word token which seems to disturb behavior. As said earlier, zero-shot generalization on Type 5 cannot do better than 0.5, as it cannot infer that plants only require water.

*Policy n-shot generalization.* When goal imagination begins (Figures 2c and 2d after the vertical line), agents can imagine goals and train on them. This means that $\overline{\text{SR}}$ evaluates n-shot policy generalization.

Agents can now perform *behavior adaptation*. They can learn that plants need water. As they learn this, their generalization performance on goals from Type 5 increases and goes beyond 0.5. Note that this effects fights the zero-shot generalization. By default, policy and reward function apply zero-shot generalization: e.g. they bring water or food equally to plants. Behavioral adaptation attempts to modify that default behavior. Because of the poor zero-shot generalization of the reward on goals of Type 2, agents cannot hope to learn Type 2 behaviors. Moreover, Type 2 goals cannot be imagined, as the word *flower* is unknown to the agent.

(a) **Reward Function, no imagination**

(b) **Policy, no imagination**

(c) **Policy, imagination half way**

(d) **Policy, imagination early**

Figure 2: **Zero-shot and n-shot generalizations of the reward function and policy.** Each figure represents the training and testing performances (split by generalization type) for the reward (a), and the policy (b, c, d). (a) and (b) represent zero-shot performance in the *no imagination* conditions. In (c) and (d), agents start to imagine goals as denoted by the vertical dashed line. Before that line, $\overline{SR}$ evaluate zero-shot generalization. After, it evaluates the n-shot generalization, as agent can train autonomously on imagined goals.

# 3    Focus on exploration

**Interesting Interactions.**    *Interesting interactions* are trajectories of the agent that humans could infer as goal-directed. If an agent brings water to a plant and grows it, it makes sense for a human. If it then tries to do this for a lamp, it also feels goal-directed, even though it does not work. This type of behavior characterizes the penchant of agents to interact with objects around them, to try new things and, as a result, is a good measure of exploration.

**Sets of interesting interactions.**    We consider three sets of interactions: 1) interactions related to training goals; 2) to testing goals; 3) the extra set. This *extra set* contains interactions where the agent brings water or food to a piece of furniture or to another supply. Although such behaviors do not achieve any of the goals, we consider them as interesting exploratory behaviors. Indeed, they testify that agents try to achieve imagined goals that are meaningful from the point of view of an agent that does not already know that doors cannot be grown, i.e. corresponding to a meaningful form of generalization after discovering that animals or plants can be grown (e.g. *grow any door*).

**The Interesting Interaction Count metric.**    We count the number of interesting interactions computed over all final transitions from the last $600$ episodes (1 epoch). Agents do not need to target these interactions, we just report the number of times they are experienced. Indeed, the agent does not have to target a particular interaction for the trajectory to be interesting from an exploratory point of view. The HER mechanism ensures that these trajectories can be replayed to learn about any goal, imagined or not. Computed on the extra set, the *Interesting Interaction Count* (I2C) is the number of times the agent was found to bring supplies to a furniture or to other supplies over the last epoch:

$$\text{I2C}_{\text{extra}} = \sum_{i \in \mathcal{I} = \mathcal{G}_{\text{extra}}} \sum_{t=1}^{600} \delta_{i,t},$$

where $\delta_{i,t} = 1$ if interaction $i$ was achieved in episode $t$, 0 otherwise and $\mathcal{I}$ is the set of interesting interactions (here from the extra set) performed during an epoch.

Agents that are allowed to imagine goals achieve higher scores in the testing and extra sets of interactions, while maintaining similar exploration scores on the training set, see Figures 3a to 3c.

(a)          (b)          (c)

Figure 3: **Exploration metrics** (a) Interesting interaction count (I2C) on training set, (b) I2C on testing set, (c) I2C on extra set. Goal imagination starts early (vertical blue line), half-way (vertical orange line) or does not start (*no imagination* baseline in green).

# 4 Focus on Goal Imagination

Algorithm 2 presents the algorithm underlying our goal imagination mechanism. This mechanism is inspired from the *Construction Grammar* (CG) literature and generates new sentences by composing known ones [7]. It computes sets of equivalent words by searching for sentences with an edit distance of 1: sentences where only one word differs. These words are then labelled equivalent, and can be substituted in known sentences. Note that the goal imagination process filters goals that are already known. Although all sentences from $\mathcal{G}^{\text{train}}$ can be imagined, there are filtered out of the imagined goals as they are discovered. Imagining goals from $\mathcal{G}^{\text{train}}$ before they are discovered drives the exploration of IMAGINE agents. In our setup, however, this effect remains marginal as all the goals from $\mathcal{G}^{\text{train}}$ are discovered in the first epochs (see Figure 5).

**Algorithm 2**: Goal Imagination. The edit distance between two sentences refers to the number of words to modify to transform one sentence into the other.

1: **Input:** $\mathcal{G}_{\text{known}}$ (discovered goals)
2: **Initialize:** *word_eq* (list of sets of equivalent words, empty)
3: **Initialize:** *goal_template* (list of template sentences used for imagining goals, empty)
4: **Initialize:** $\mathcal{G}_{\text{im}}$ (empty)
5: **for** $g_{\text{NL}}$ in $\mathcal{G}_{\text{known}}$ **do** {Computing word equivalences}
6:    *new_goal_template* = True
7:    **for** $g_m$ in *goal_template* **do**
8:      **if** edit_distance($g_{\text{NL}}, g_m$) $< 2$ **then**
9:        *new_goal_template* = False
10:      **if** edit_distance($g_{\text{NL}}, g_m$) $== 1$ **then**
11:        $w_1, w_2 \leftarrow$ *get_non_matching_words*($g_{NL}, g_m$)

12:        **if** $w_1$ and $w_2$ not in any of *word_eq* sets **then**
13:          *word_eq*.add($\{w_1, w_2\}$)
14:        **else**
15:          **for** *eq_set* in *word_eq* **do**
16:            **if** $w_1 \in eq\_set$ or $w_2 \in eq\_set$ **then**
17:              $eq\_set = eq\_set \cup \{w_1, w_2\}$
18:    **if** *new_goal_template* **then**
19:      *goal_template*.add($g_{\text{NL}}$)
20: **for** $g$ in *goal_template* **do** {Generating new sentences}
21:    **for** $w$ in $g$ **do**
22:      **for** *eq_set* in *word_eq* **do**
23:        **if** $w \in eq\_set$ **then**
24:          **for** $w'$ in *eq_set* **do**
25:            $g_{im} \leftarrow$ replace($g, w, w'$)
26:            **if** $g_{im} \notin \mathcal{G}_{\text{known}}$ **then**
27:              $\mathcal{G}_{\text{im}} = \mathcal{G}_{\text{im}} \cup \{g_{im}\}$
28: $\mathcal{G}_{\text{im}} = \mathcal{G}_{\text{im}} \setminus \mathcal{G}_{\text{known}}$       {filtering known goals.}

Figure 4: Venn diagram of goal spaces.

Table 2: All imaginable goals $\mathcal{G}^{\text{im}}$ generated by the Construction Grammar Heuristic.

| | |
|---|---|
| Goals from $\mathcal{G}^{\text{train}}$ | $\mathcal{G}^{\text{train}}$. (Note that known goals are filtered from the set of imagined goals. However, any goal from $\mathcal{G}^{\text{train}}$ can be imagined before it is encountered in the interaction with SP.) |
| Goals from $\mathcal{G}^{\text{test}}$ | All goals from Type 1, 3, 4 and 5, see Table 1 |
| Syntactically incorrect goals | *Go bottom top*, *Go left right*, *Grasp red blue thing*, *Grow blue red thing*, *Go right left*, *Go top bottom*, *Grasp green blue thing*, *Grow green red thing*, *Grasp green red thing* *Grasp blue green thing*, *Grasp blue red thing*, *Grasp red green thing*. |
| Syntactically correct but unachievable goals | *Go center bottom*, *Go center top*, *Go right center*, *Go right bottom*, *Go right top*, *Go left center*, *Go left bottom*, *Go left top*, *Grow green cupboard*, *Grow green sink*, *Grow blue lamp*, *Go center right*, *Grow green window*, *Grow blue carpet*, *Grow red supply*, *Grow any sofa*, *Grow red sink*, *Grow any chair*, *Go top center*, *Grow blue table*, *Grow any door*, *Grow any lamp*, *Grow blue sink*, *Go bottom center*, *Grow blue door*, *Grow blue supply*, *Grow green carpet*, *Grow blue furniture*, *Grow green supply*, *Grow any window*, *Grow any carpet*, *Grow green furniture*, *Grow green chair*, *Grow green food*, *Grow any cupboard*, *Grow red food*, *Grow any table*, *Grow red lamp* , *Grow red door*, *Grow any food*, *Grow blue window*, *Grow green sofa*, *Grow blue sofa*, *Grow blue desk*, *Grow any sink*, *Grow red cupboard*, *Grow green door*, *Grow red furniture*, *Grow blue food*, *Grow red desk* , *Grow red table*, *Grow blue chair*, *Grow red sofa*, *Grow any furniture*, *Grow red window*, *Grow any desk*, *Grow blue cupboard*, *Grow red chair*, *Grow green desk*, *Grow green table*, *Grow red carpet*, *Go center left*, *Grow any supply*, *Grow green lamp*, *Grow blue water*, *Grow red water*, *Grow any water*, Grow green water, *Grow any water*, Grow green water. |

**Imagined goals.** We run our goal imagination mechanism based on the Construction Grammar Heuristic (CGH) from $\mathcal{G}^{\text{train}}$. After filtering goals from $\mathcal{G}^{\text{train}}$, this produces 136 new imagined sentences. Table 2 presents the list of these goals while Figure 4 presents a Venn diagram of the various goal sets. Among these 136 goals, 56 belong to the testing set $\mathcal{G}^{\text{test}}$. This results in a coverage of 87.5% of $\mathcal{G}^{\text{test}}$, and a precision of 45%. In goals that do not belong to $\mathcal{G}^{\text{test}}$, goals of the form *Grow* + {*any*} ∪ **color** + **furniture** ∪ **supplies** (e.g. *Grow any lamp*) are *meaningful* to humans, but are not achievable in the environment (*impossible*).

**Variants of goal imagination mechanisms.** Main Section 4.2 investigates variants of our goal imagination mechanisms:

1. *Lower coverage:* To reduce the coverage of CGH while maintaining the same precision, we simply filter half of the goals that would have been imagined by CGH. This filtering is probabilistic, resulting in different imagined sets for different runs. It happens online, meaning that the coverage is always half of the coverage that CGH would have had at the same time of training.

2. *Lower precision:* To reduce precision while maintaining the same coverage, we sample a random sentence (random words from the words of $\mathcal{G}^{\text{train}}$) for each goal imagined by CGH that does not belong to $\mathcal{G}^{\text{test}}$. Goals from $\mathcal{G}^{\text{test}}$ are still imagined via the CGH mechanism. This variants only doubles the imagination of sentences that do not belong to $\mathcal{G}^{\text{test}}$.

3. *Oracle:* Perfect precision and coverage is achieved by filtering the output of CGH, keeping only goals from $\mathcal{G}^{\text{test}}$. Once the 56 goals that CGH can imagine are imagined, the oracle variants adds the 8 remaining goals: those including the word *flower* (Type 2 generalization).

4. *Random goals:* Each time CGH would have imagined a new goal, it is replaced by a randomly generated sentence, using words from the words of $\mathcal{G}^{\text{train}}$.

Note that all variants imagine goals at the same speed as the CGH algorithm. They simply filter or add noise to its output, see Figure 5.

Figure 5: **Evolution of known goals for various goal imagination mechanisms.** All graphs show the evolution of the number of goals from $\mathcal{G}^{\text{train}}$, $\mathcal{G}^{\text{test}}$ and others in the list of known goals $\mathcal{G}_{\text{known}}$. We zoom on the first epochs, as most goals are discovered and invented early. Vertical dashed line indicates the onset of goal imagination. (a) CGH; (b) Low Coverage; (c) Low precision; (d) Oracle; (e) Random Goals.

**Effect of low coverage on generalization.** In Main Section 4.2, we compare our goal imagination mechanism to a *Low Coverage* variant that only covers half of the proportion of $\mathcal{G}^{\text{test}}$ covered by CGH (44%). Figure 6 shows that the generalization performance on goals from $\mathcal{G}^{\text{test}}$ that the agent imagined (n-shot generalization, blue) are not significantly higher than the generalization performance on goals from $\mathcal{G}^{\text{test}}$ that were not imagined (zero-shot generalization). As they are both significantly higher than the *no imagination* baseline, this implies that training on imagined goals boosts zero-shot generalization on similar goals that were not imagined.

Figure 6: **Zero-shot versus n-shot.** We look at the *Low Coverage* variant of our goal imagination mechanism that only covers 43.7% the test set with a 45% precision. We report success rates on testing goals of Type 5 (*grow + plant*) and compare with the *no imagination* baseline (green). We split in two: goals that were imagined (blue), and goals that were not (orange).

**Details on the impacts of various goal imagination mechanisms on exploration.** Figure 7 presents the I2C exploration scores on the training, testing and extra sets for the different goal imagination mechanisms introduced in Main Section 4.2. Let us discuss each of these scores:

1. *Training interactions.* In Figure 7a, we see that decreasing the precision (Low Precision and Random Goal conditions) affects exploration on interactions from the training set, where it falls below the exploration of the *no imagination* baseline. This is due to the addition of meaningless goals forcing agent to allow less time to meaningful interactions relatively.

2. *Testing interactions.* In Figure 7b, we see that the highest exploration scores on interactions from the test set comes from the oracle. Because it shows high coverage and precision, its spends more time on the diversity of interactions from the testing set. What is more surprising is the exploration score of the low coverage condition, higher than the exploration score of CGH. With an equal precision, CGH should show better exploration, as it covers more test goals. However, the *Low Coverage* condition, by spending more time exploring each of its imagined goals (it imagined fewer), probably learned to master them better, increasing the robustness of its behavior towards those. This insight advocates for the use of goal selection methods based on learning progress [6, 4]. Agents could estimate their learning progress on imagined goals using their internal reward function and its zero-shot generalization. Focusing on goals associated to high learning progress might help agents filter goals they can learn about from others.

3. *Extra interactions.* Figure 7c shows that only the goal imagination mechanisms that invent goals not covered by the testing set manage to boost exploration in this extra set. The oracle perfectly covers the testing set, but does not generate goals related to other objects (e.g. *grow any lamp*).

(a) I2C on $\mathcal{G}^{\text{train}}$      (b) I2C on $\mathcal{G}^{\text{test}}$      (c) I2C on $\mathcal{G}^{\text{extra}}$

Figure 7: **Exploration metrics for different goal imagination mechanisms**: (a) Interesting interaction count (I2C) on training set, I2C on testing set, (c) I2C on extra set. Goal imagination starts early (vertical line), except for the *no imagination* baseline (green). Standard errors of the mean plotted for clarity (as usual, 10 seeds).

# 5 Focus on Architectures

This section compares our proposed object-based modular architecture MA for the policy and reward function to a flat architecture that does not use inductive biases for efficient skill transfer. We hypothesize that only the object-based modular architectures enable a generalization performance that is sufficient for the goal imagination to have an impact on generalization and exploration. Indeed, when generalization abilities are low, agents cannot evaluate their performance on imagined goals and thus, cannot improve.

**Preliminary study of the reward function architecture.** We first compared the use of modular and flat architectures for the reward function (MA$^\mathcal{R}$ vs FA$^\mathcal{R}$ in Figure 8). This experiment was conducted independently from policy learning, in a supervised setting. We use a dataset of $50 \times 10^3$ trajectories and associated goal descriptions collected using a pre-trained policy. To closely match the training conditions of IMAGINE, we train the reward function on the final states $s_T$ and test it on any states $s_t, t = [1, .., T]$ of other episodes. Table 3 provides the $F_1$ score computed at convergence on $\mathcal{G}^{\text{train}}$ and $\mathcal{G}^{\text{test}}$ for the two architectures.

Table 3: Reward function architectures performance.

|  | $F_{1\,\text{train}}$ | $F_{1\,\text{test}}$ |
|---|---|---|
| MA$^\mathcal{R}$ | $0.98 \pm 0.02$ | $0.64 \pm 0.22$ |
| FA$^\mathcal{R}$ | $0.60 \pm 0.10$ | $0.22 \pm 0.05$ |

MA$^\mathcal{R}$ outperforms FA$^\mathcal{R}$ on both the training and testing sets. In addition to its poor generalization performance, FA$^\mathcal{R}$'s performance on the training set are too low to support policy learning. As a result, the remaining experiments in this paper use the MA$^\mathcal{R}$ architecture for all reward functions. Thereafter, MA is always used for the reward function and the terms MA and FA refer to the architecture of the policy.

**Architectures representations.** The combination of MA for the reward function and either MA or FA for the policy are represented in Figure 9.

**Policy architecture comparison.** Table 4 shows that MA significantly outperforms FA on both the training and testing sets at convergence. Figure 10a clearly shows an important gap between the generalization performance of the modular and the flat architecture. In average, less than 20% of the testing goals can be achieved with FA when MA masters half of them without imagination. Moreover, there is no significant difference between the never and the early imagination conditions for the flat architecture. The generalization boost enabled by the imagination is only observable for the modular architecture (see Main Table 1). Figure 10c and 10d support similar conclusions for exploration: only the modular architecture enable goal imagination to drive an exploration boost on the testing and extra sets of interactions.

Table 4: Architectures performance. Both p-values $< 10^{-10}$.

|  | $\overline{\text{SR}}_{\text{train}}$ | $\overline{\text{SR}}_{\text{test}}$ |
|---|---|---|
| MA | $0.95 \pm 0.05$ | $0.76 \pm 0.10$ |
| FA | $0.40 \pm 0.13$ | $0.16 \pm 0.06$ |

In preliminary experiments, we tested a *Flat-Concatenation* (FC) architecture where the gated attention mechanism was replaced by a simple concatenation of goal encoding to the state vector. We did not found signficant difference with respect to FA. We chose to pursue with the attention mechanism, as it improves model interpretability (see Additional Visualization 7).

Figure 8: **Reward function architectures**: (a) *Flat-attention* reward function (FA$^{\mathcal{R}}$) and (b) *Modular-attention* reward function (MA$^{\mathcal{R}}$). We use MA$^{\mathcal{R}}$ for all experiments except for the experiment in Table 3

Figure 9: **Policy and reward function architectures:** (a) *Modular-attention* (MA) reward + *Flat-attention* (FA) policy. (b) MA reward + MA policy. In both figures, the reward function is represented on the right in green, the policy on the left in pink, the language encoder in the bottom in yellow and the attention mechanisms at the center in blue.

Figure 10: **Policy architecture comparison:** (a) $\overline{\mathrm{SR}}$ on $\mathcal{G}^{\mathrm{test}}$ for the FA and MA architectures when the agent starts imagining goals early (plain, after the black vertical dashed line) or never (dashed). (b, c, d) I2C on interactions from the training, testing and extra sets respectively. Imagination is performed using CGH. Stars indicate significant differences between CGH and the corresponding *no imagination* baseline.

# 6 Focus on Reward Function

Our IMAGINE agent is autonomous and, as such, needs to learn its own reward function. It does so by leveraging a weak supervision from a social partner that provides descriptions in a simplified language. This reward function can be used for many purposes in the architecture. This paper leverages some of these ideas (the first two), while others are left for future work (the last two):

- **Behavior Adaptation.** As Main Section 4.1 showed, the reward function enables agents to adapt their behavior with respect to imagined goals. Whereas the zero-shot generalization pushed agents to grow plants with food and water with equal probability, the reward function helped agents to correct that behavior towards more water.

- **Guiding Hindsight Experience Replay (HER).** In multi-goal RL with discrete sets of goals, HER is traditionally used to modify transitions sampled from the replay buffer. It replaces originally targeted goals by others randomly selected from the set of goals [1, 11]. This enables to transfer knowledge between goals, reinterpreting trajectories in the light of new goals. In that case, a reward function is required to compute the reward associated to that new transition (new goal). To improve on random goal replay, we favor goal substitution towards goals that actually match the state and have higher chance of leading to rewards. In IMAGINE, we scan a set of 40 goal candidates for each transition, and select substitute goals that match the scene when possible, with probability $p = 0.5$.

- **Exploring like Go-Explore.** In Go-Explore [5], agents first reach a goal state, then start exploring from there. We could reproduce that behavior in our IMAGINE agents with our internal reward function. The reward function would scan each state during the trajectory. When the targeted goal is found to be reached, the agent could switch to another goal, add noise on its goal embedding, or increase the exploration noise on actions. This might enable agents to explore sequences of goal-directed behaviors. We leave the study of this mechanism for future work.

- **Filtering of Imagined Goals.** When generating imagined goals, agents also generate meaningless goals. Ideally, we would like agents to filter these from meaningful goals. Meaningful goals, are goals the agent can interpret with its reward function, goals from which it can learn directed behavior. They are interpreted from known related goals via the generalization of the reward function. If we consider an ensemble of reward functions, chances are that all reward functions in the ensemble will agree on the interpretation of meaningful imagined goals. On the other hand, they might disagree on meaningless goals, as their meanings might not be as easily derived from known related goals. Using an ensemble of reward function may thus help agents filter meaningful goals from meaningless ones. This could be done by labeling a dataset of trajectories with positive or negative rewards and comparing results between reward functions, effectively computing agreement measures for each imagined goals. Having an efficient filtering mechanism would drastically improve the efficiency of goal imagination, as Main Section 4.2 showed that the ratio of meaningful goals determines generalizations performance. This is also left for future work.

# 7 Additional Visualizations

**Visualizing Goal Embedding**    To analyze the goal embeddings learned by the language encoder $L_e$, we perform a t-SNE using 2 components, perplexity 20, a learning rate of 10 for 5000 iterations. Figure 11 presents the resulting projection for a particular run. The embedding seems to be organized mainly in terms of motor predicates (11a), then in terms of colors (11b). Object types or categories do not seem to be strongly represented (11c).

Figure 11: **t-SNE of Goal Embedding.** The same t-SNE is presented, with different color codes (a) predicates, (b) colors, (c) object categories.

**Visualizing Attention Vectors**    In the *modular-attention* architectures for the reward function and policy, we train attention vectors to be combined with object-specific features using a gated attention mechanism. In each architecture, the attention vector is shared across objects (permutation invariance). Figure 12 presents examples of attention vectors for the reward function (12a) and for the policy (12b) at the end of training. These attention vectors highlight relevant parts of the object-specific sub-state depending on the NL goal:

- When the sentence refers to a particular object type (e.g. *dog*) or category (e.g. *living thing*), the attention vector suppresses the corresponding object type(s) and highlights the complement set of object types. If the object does not match the object type or category described in the sentence, the output of the Hadamard product between object types and attention will be close to 1. Conversely, if the object is of the required type, the attention suppression ensures that the output stays close to zero. Although it might not be intuitive for humans, it efficiently detects whether the considered object is the one the sentence refers to.

- When the sentence refers to a navigation goal (e.g. *go top*, the attention highlights the agent's position (here $y$).

- When the sentence is a *grow* goal, the reward function focuses on the difference in object's size, while the policy further highlights the object's position.

The attention vectors uses information about the goal to highlight or suppress parts of the input using the different strategies described above depending on the type of input (object categories, agent's position, difference in size etc). This type of gated-attention improves the interpretability of the reward function and policy.

Figure 12: **Attention vectors** (a) $\boldsymbol{\alpha}^g$ for the reward function (1 seed). (b) $\boldsymbol{\beta}^g$ for the policy (1 seed).

# 8   Comparing IMAGINE to goal-as-state approaches.

In the goal-conditioned RL literature, some works have proposed goal generation mechanisms to facilitate the acquisition of skills over large sets of goals [13, 15, 4, 12]. Some of them had a special interest in exploration, and proposed to bias goal sampling towards goals from low density areas [15]. One might then think that IMAGINE should be compared to these approaches. However, there are a few catches:

1. Nair et al. [13, 12], Pong et al. [15] use generative models of states to sample state-based goals. However, our environment is procedurally generated. This means that sampling a given state from the generative model has a very low probability to *match* the scene. If the present objects are three red cats, the agent has no chance to reach a goal specifying dogs and lions' positions, colors and sizes. Indeed, most of the state space is made of object features that cannot be acted upon (colors, types, sizes of most objects). One could imagine using SP to organize the scene, but we would need to ask SP to find the three objects specified by the generated goal, in the exact colors (RGB codes) and size. Doing so, there would be no distracting object for agent to discover and learn about. A second option is to condition the goal generation on the scene as it is done in Nair et al. [12]. The question of whether it might work in procedurally-generated environments remains open.

2. Assuming a perfect goal generator that only samples valid goals that do not ask a change of object color or type, the agent would then need to bring each object to its target position and to grow objects to their very specific goal size. These goals are not the same as those targeted by IMAGINE, they are too specific. These approaches –like most goal-conditioned RL approaches– represent goals as particular states (e.g. block positions in manipulation tasks, visual states in navigation tasks) [16, 1, 13, 15, 4]. In contrast, language-conditioned agents represent abstract goals, usually defined by specific constraints on states (e.g. *grow any plant* requires the size of at least one plant to increase) [2, 9, 3]. For this reason, *goal-as-state* and *abstract goal* approaches do not tackle the same problem. The first targets specific coordinates, and cannot be instructed to reach abstract goals, while the second are not trained to reach specific states.

For these reasons, we argue that the goal-conditioned approaches that use state-based goals cannot be easily or fairly compared to our approach IMAGINE.

# 9 Implementation details

**Reward function inputs and hyperparameters.** Supplementary Section 5 details the architecture of the reward function. The following provides extra details about the inputs. The object-dependent sub-state $\mathbf{s}_{obj(i)}$ contains information about both the agent's body and the corresponding object $i$: $\mathbf{s}_{obj(i)} = [\mathbf{o}_{body}, \Delta\mathbf{o}_{body}, \mathbf{o}_{obj(i)}, \Delta\mathbf{o}_{obj(i)}]$ where $\mathbf{o}_{body}$ and $\mathbf{o}_{obj(i)}$ are body- and $obj_i$-dependent observations, and $\Delta\mathbf{o}_{body}^t = \mathbf{o}_{body}^t - \mathbf{o}_{body}^0$ and $\Delta\mathbf{o}_{obj(i)}^t = \mathbf{o}_{obj(i)}^t - \mathbf{o}_{obj(i)}^0$ measure the difference between the initial and current observations. The second input is the attention vector $\boldsymbol{\alpha}^g$ that is integrated with $\mathbf{s}_{obj(i)}$ through an Hadamard product to form the model input: $\mathbf{x}_i^g = \mathbf{s}_{obj(i)} \odot \boldsymbol{\alpha}^g$. This attention vector is a simple mapping from $\mathbf{g}$ to a vector of the size of $\mathbf{s}_{obj(i)}$ contained in $[0,1]^{size(\mathbf{s}_{obj(i)})}$. This cast is implemented by a one-layer neural network with sigmoid activations $\text{NN}^{cast}$ such that $\boldsymbol{\alpha}^g = \text{NN}^{cast}(\mathbf{g})$.

For the three architectures the number of hidden units of the LSTM and the sizes of the hidden layers of fully connected networks are fixed to 100. NN parameters are initialized using He initialization [8] and we use one-hot word encodings. The LSTM is implemented using `rnn.BasicLSTMCell` from tensorflow 1.15 based on Zaremba et al. [17]. The states are initially set to zero. The LSTM's weights are initialized uniformly from $[-0.1, 0.1]$ and the biases initially set to zero. The LSTM use a $tanh$ activation function whereas the NN are using ReLU activation functions in their hidden layers and sigmoids at there output.

**Reward function training schedule.** The architecture are trained via backpropagation using the Adam Optimizer [10]. The data is fed to the model in batches of 512 examples. Each batch is constructed so that it contains at least one instance of each goal description $g_{\text{NL}}$ (goals discovered so far). We also use a modular buffer to impose a ratio of positive rewards of 0.2 for each description in each batch. When trained in parallel of the policy, the reward function is updated once every 1200 episodes. Each update corresponds to up to 100 training epochs (100 batches). We implement a stopping criterion based on the $F_1$-score computed from a held-out test set uniformly sampled from the last episodes (20% of the last 1200 episodes (2 epochs)). The update is stopped when the $F_1$-score on the held-out set does not improve for 10 consecutive training epochs.

**RL implementation and hyperparameters.** In the policy and critic architectures, we use hidden layers of size 256 and ReLU activations. Attention vectors are cast from goal embeddings using single-layer neural networks with sigmoid activations. We use the He initialization scheme for [8] and train them via backpropagation using the Adam optimizer ($\beta_1 = 0.9, \beta_2 = 0.999$) [10].

Our learning algorithm is built on top of the OpenAI Baselines implementation of HER-DDPG.[1] We leverage a parallel implementation with 6 actors. Actors share the same policy and critic parameters but maintain their own memory and conduct their own updates independently. Updates are then summed to compute the next set of parameters broadcast to all actors. Each actor is updated for 50 epochs with batches of size 256 every 2 episodes of environment interactions. Using hindsight replay, we enforce a ratio $p = 0.5$ of transitions associated with positive rewards in each batch. We use the same hyperparameters as Plappert et al. [14].

**Computing resources.** The RL experiments contain 8 conditions of 10 seeds each, and 4 conditions with 5 seeds (SP study). Each run leverages 6 cpus (6 actors) for about 36h for a total of 2.5 cpu years. Experiments presented in this paper requires machines with at least 6 cpu cores.

## Footnotes

[1]The OpenAI Baselines implementation of HER-DDPG can be found at https://github.com/openai/baselines, our implementation can be found at `https://sites.google.com/view/imagine-drl`.[2]