[Reviews · NeurIPS 2020]

Review 1

Summary and Contributions: This paper is interested in exploring the idea that language, if used to set goals, can be used to exploration the environment. Unlike the setting typically considered in instruction following literature, here the language grounding is learned mainly by having access to a social partner (SP) which provides language goal descriptions of terminal states reached by the agent. These goals are added to the growing set of goals the agent is trained to reach and later expanded with out-of-distribution generated goals (all language is generated from a simple grammar with 256 possible sentences). The resulting goal-conditional agents are evaluated with respect to generalization to unseen goals and whether they exhibit goal-directed behaviors. The paper also proposes an architecture that aids systematic generalization and generalization to unseen states.

Strengths: The question of how could language structure and aid exploration in RL is interesting and underexplored. The proposed architecture seems promising, the generalization improvement is significant against the flat baseline.

Weaknesses: The motivation for studying the proposed setting is unclear to me. Could you provide some real-world examples of RL problems, where a human is available to label accessed goals but not feedback on whether a set goal has been reached? As it is explained now, in terms of insights gained, it seems to be comparable to settings considered in instruction following literature.

Correctness: The paper could be significantly improved if the learned behaviors were evaluated in a transfer learning setting, i.e. by using the learned policies on some related downstream tasks. As far as I can tell goal imagination helps with generalization to unseen goals, because learning a reward model that generalizes is easier than learning a policy that generalizes, hence training by adding more goals to follow helps with the generalization of policies.

Clarity: The paper is mostly clearly written. Given the key ideas are explained there, the clarity of exposition in section 3.2 could be improved. When possible, e.g. adding a function form of each module, include an algorithm box.

Relation to Prior Work: When introducing the new environment, there should be some discussion and comparison to the existing gridworld environments for instruction following. Why was The Playground environment developed, what can be studied with it that can not be studied with e.g. BabyAI?

Reproducibility: Yes

Additional Feedback: A more thorough examination of the proposed architecture could be valuable, in particular, comparing it to other proposed architectures shown to help systematic generalization (Bahdanau et al, 2018) Section 5 should be split into Conclusion and Future Work.


Review 2

Summary and Contributions: This paper studies intrinsicly motivated artificial agents in the context of natural-language-based imagination of potential goal states. To this end, this work introduces a new "Playground" environment, a 2D canvas on which the agent moves a hand to interact with a subset of possible items. Within "Playground," a templated language is used to specify goals (e.g. "grasp any flower") and the agent is tasked with exploring this environment to discover (training) goals which can then be extrapolated (via "imagination") to new potential goals. The authors present a neural "IMAGINE" architecture and training pipeline for this intrinsically motivated task. Trained agents are then evaluated based on their ability to generalize to new (test set) goals. These evaluations show that the capacity to imagine new natural-language goals during training is critical to generalization. #### POST-REBUTTAL RESPONSE #### This paper has presented an novel methodological contribution and provided extensive evaluation on a new environment. While I would very much like to see these ideas extended to, and evaluated on, a new domain (e.g. BabyAI) I don't believe such an extension is a necessary component of this work. I am very happy to see that the authors, in the rebuttal, have committed to open sourcing their environment, I look forward to others using this in the future. Altogether I see no reason to change my score, this paper remains a clear accept to me.

Strengths: This paper makes several strong contributions, the empirical evaluations are *extremely* thorough, and code is provided to reproduce and build upon their work. In particular, I would highlight the following strengths: 1. The idea of hallucinating goals with natural-language to improve exploration and generalization is interesting and appears novel. 2. The Playground environment is a nice test-bed for further study in this area. 3. The paper has established strong, and very well evaluated, baselines.

Weaknesses: This paper has chosen to simplify and distill the environment/task so as to be able to very carefully analyze their results. Some of these simplifications include: 1. The number of variety of interactions enabled by Playground is very limited and the number of possible goals (256) is quite small. While the language used to specify goals is technically "natural," it is very constrained. 2. Several strong assumptions are made regarding the abilities of the social partner agent (who, post-hoc, describes what an agent has accomplished during exploration) in Section 2 (admittedly, the results of relaxing two of these assumptions are evaluated). The weakness of this approach distillation approach is that it is unclear if the presented ideas can be used for other RL tasks of interest to the community. The authors here cite the BabyAI environment, experiments showing if any of the proposed ideas transfer to this environment would be very interesting and help answer the question: should I attempt to adapt IMAGINE to my setting of interest?

Correctness: Yes, the claims and empirical methodology appear correct.

Clarity: The paper is well written although (in the additional feedback section of this review) I provide some line-by-line comments that I believe can further improve clarity.

Relation to Prior Work: The paper seems well positioned against prior work although I would have liked to see some discussion of instruction following in more complicated (visual) environments as it is in such environments that I believe this type of imagination might have a substantial impact. Some seed papers include https://arxiv.org/abs/1912.01734 and https://arxiv.org/abs/1812.04155.

Reproducibility: Yes

Additional Feedback: Line-by-line comments: Fig. 1 - This figure is quite cluttered. I would recommend removing/simplifying some of the graphics (e.g. the thought bubbles) and, when possible, moving them outside the environment canvas. Line 37 - Do the citations on lines 38-39 substantiate the preceding claim that language actually influences children's exploration behavior? This seems like a very difficult claim to test, how do you disentangle mental maturity with language acquisition? Note that I do not consider children "narrating their ongoing activities" to be a meaningful change in behavior if it is not accompanied by a change in how the children actually complete those activities. Line 57 - Including BabyAI as a reference in this line seems a bit awkward. Would it not better fit in the citations on line 61? Line 61 - As discussed above, a discussion of embodied instruction following would be interesting here. Line 120 - Why not evaluate the effect of relaxing the "precision" assumption as well? Line 136 - It seems somewhat strange to use the same font when referencing metrics (e.g. SR) as when referencing other components (e.g. SP). Line 143 - Rather than "Welch's t-test" I would say "unequal variances t-test" as it is more descriptive. Moreover it's somewhat odd to reference a test without a discussion of the null hypothesis. I also assume you are evaluating the models on the same set of test goals/environment configurations, if this is the case why not pair the data and use a paired t-test? Line 151 - How easy is it to extend Playground to include new objects and interactions? Line 159 - Please also mention the total number of possible (not necessarily sensible) utterances in your language. Lines 167-169 - It would be great to know the sizes of the testing and training goal sets here. Line 176 - Why use a capital pi here, I'm used to seeing lower case pi used for policies. Line 178 - It would be interesting to clarify what happens here if the imagined goal is impossible, how is the environment set up in light of lines 108-113? Lines 182 - I could use a reminder here that G(s_T) is an exhaustive enumeration of compatible goals within the training set. Lines 186-188 - This description makes it seem as though you only ever add experiences for which the goal is within the training set. From my understanding of the supplementary results, this is not the case and you do enqueue imagined goals with their predicted rewards, correct? Lines 190-191 - Did you experiment with other sampling strategies? It would seem reasonable to me to sample so that unsampled goals are more frequently selected. Equation after line 230 - This equation is completely unconnected from the text. Even displayed equations should be treated as though they are inline. Fig. 4 - Is it not surprising that the lower coverage heuristic results in a larger I2C? - Rather than reporting this as the SEM perhaps report it as an X% CI?


Review 3

Summary and Contributions: The authors present an algorithm for proposing unseen goals by leveraging the compositionality of natural language. Some goals are provided by a social peer to define the goal space. The agent then proposes unseen goals with a learned model of the sentence grammar, labels past trajectories with learned hindsight rewards for imagined goals, and uses these trajectories to learn the RL agent. The learned hindsight reward and RL agent learn object representations via set attention.

Strengths: - The paper studies an interesting aspect of human learning that is underexplored in the reinforcement learning community. - Experiments show that compositional goal generalization facilitates both exploration and task generalization and performs competitively with other ways of proposing goals. - The paper is well written and the method is described clearly.

Weaknesses: - The presented method contains many components and further ablations would be interesting, for example on the choice of object representations. - The introduced environment is quite simple, allowing for only 256 different sentences. It would be interesting to scale the approach to more complex tasks that necessitate representation learning, similar to prior works on natural-language-instructed RL agents in the Doom and DMLab environments.

Correctness: - I think claiming the use of language compositionality for exploration is too large of a claim. For example, see "Language as an abstraction for hierarchical deep reinforcement learning" (NeurIPS 2019). - Section 2 paragraph headline "Learning objective" -> should this be "evaluation metrics" instead?

Clarity: - I think the terminology of "out-of-distribution goal" is a bit unclear and suggests a technical meaning that is probably not intended. Otherwise, it would require defining the distribution. For the empirical distribution of past experience, any unseen goal is out-of-distribution. For a good model of past experience (e.g. a language model), new combinations of old factors can be in-distribution. What is really mean might be generalization, such that interesting goals are in-distribution. - The text in Fig 1 is too small to read.

Relation to Prior Work: - Related work is adequately discussed.

Reproducibility: Yes

Additional Feedback:


Review 4

Summary and Contributions: In this paper, the author introduce a learning schema IMAGINE(intrinsic motivations and goal invention for exploration). It is an intrinsically motivated deep reinforcement agent using goal imagination(the ability to imagine out of distribution goal) to interact with the environment by leveraging natural language description from social partners. This framework uses the concept of imagination and tries to solve open-ended learning environments. The key contribution of the paper is, given the goal description from the SP, the agent would be able to generate relative similar goals beyond the original distribution of goal sets based on the grammar. This would be helped for skill transferring and generalization. Authors also design an novel test environment.

Strengths: The authors present a novel method for "new goals description" and encoding "imagination". Their combined infant-parent learning dynamics and reinforcement learning. Previous work on open environment problem mostly focuses on discovering states and actions. In addition, The design of the experiment is also interesting. Based on the design, this playground can be extended to evaluate a variety of tasks. (skill transfer, language grounding, human-agent interaction etc) .

Weaknesses: 1) I think it might be hard to extend.The architecture is designed for this specific environment. Would the grammar construction process be needed for every new test setting? 2) I find some terminologies used in the paper confusing. For example, I wonder what SR (success rates) refer to and why it could captures "generalization".

Correctness: Yes

Clarity: Yes. However, I would just add a few minor suggestions: 1) The description of figure 1 is not very clear. I don’t quite understanding the meaning of "passing"/"failing" in the bottom three sub-figures. 2) Page 5, s_T was not clearly defined. Similarly, there is no claim about what is the state space s, action space a. 3) Figure 2 lacks emphasis. When I first looked at this figure, I was confused about where I should start to follow the flowchart. .

Relation to Prior Work: Yes

Reproducibility: Yes

Additional Feedback:

[Author Response · NeurIPS 2020]

**Overview.** We thank reviewers for their constructive feedback and clarified the paper based on their comments. This paper introduces language goal imagination as a new exploration mechanism, and presents a systematic study of its impact on exploration and generalization. We were pleased that reviewers acknowledged the novelty and interest of this approach. **R2** valued "the *extremely* thorough empirical evaluations", **R5** the experimental extensibility, and **R1** and **R4** the strong experimental evidence supporting our claims on exploration and generalization.

**Motivation for a descriptive setup and "real-world problems" R1.** Our setup is motivated by a developmental approach which investigates how child development can inspire the building of autonomous agents. Social interactions play a key role for children: they mostly follow their own goals and get descriptive feedback from adults (see L114, **R1**). Our descriptive setup mimics these findings, and thus differs from the classical instruction-following setup: agents set their own goals, imagine goals and receive language descriptions (*descriptive feedback*) instead of success/failure signal (*instruction feedback*). Descriptive feedback can have real-world applications. It helps sample-efficiency and facilitates human labeling with *a posteriori* counterfactual feedback (irrespective of the selected goal) and is especially useful when providing instructions is not straightforward (e.g. possible effects are unknown). We are currently working on an industrial application leveraging such descriptive setups for digital assistants interacting with real users in smart homes. It is also more user-friendly to comment the actions of an autonomous system than giving instruction that may fail. We will mention this applied work in the discussion.

**Why do we need the Playground environment? R1** Existing benchmarks (like BabyAI) were missing two features:

1. To our knowledge, existing benchmarks use instruction feedback whereas we want to study descriptive feedback.
2. In Playground, agents can ground the meaning of object categories and the meaning of category-dependent dynamics (animals grow with food or water, plants grow with water only). We can thus study new types of generalization (see Supp. Sec. 2). We believe that the full power of goal imagination occurs in environment with rich combinatorial dynamics involving a wide variety of objects.

Playground is a tool that we hope will enable the community to further study under-explored descriptive setups with rich combinatorial dynamics, as well as goal imagination (as noted by **R2** and **R5**). The environment can easily be extended by adding objects, attributes, category- or object-type-dependent dynamics (**R2**) and will be open-sourced.

**How to scale Playground to more goals? R2**, **R4** and **R5** were concerned about the size of the set of feasible goals (256) (out of $9 \cdot 1e6$ possible sentences, **R2**). However, they include different types of interactions (navigation, grasping, growing), whereas traditional approaches usually only consider navigation to a target object (e.g. Doom [12], DMLab [32]). The possible set of goals would easily grow by introducing synonyms. Using pre-trained language models would be useful there (Lynch et al., 2020). Having a small set of goals allowed us to easily control the complexity of our environment. We believe the study of new mechanisms (here goal imagination) benefits from controlled environments, rigorous methodology and statistical testing (supported by **R2**).

**How can I adapt (R2) or extend (R5) IMAGINE to other RL approaches?** Our descriptive setup was studied in isolation from instruction-feedback, but the two can be combined. Two features of IMAGINE could be added to traditional RL approaches (**R2**, **R5**). First, descriptive feedback can be used in any setup that assumes a descriptive function of the trajectory (either hard-coded [12] or learned [17]). Second, agents can imagine goals as long as they learn a reward function that can generalize their meaning from known goals ([5], as noted by **R1**). Descriptive feedback would facilitate human labeling (e.g. Lynch et al., 2020) and improve sample efficiency (via hindsight learning). Goal imagination would help with generalization and exploration as demonstrated by our experiments.

**R5** wondered whether the grammar construction process would be needed in a new setting. In new environments, reusing learned patterns is likely to boost further exploration: this suggests interesting future work. In more complex language settings, [1] showed how similar mechanisms yield fruitful results for data augmentation on NLP datasets. More realistic language could also be mapped to simpler language to facilitate goal imagination (Andreas et al., 2020).

**Other comments.** In our approach, sentences are goals. Testing goals are built from the same atomic words (null atomic divergence), but have maximum compound divergence [41]: they are *out-of-distribution* w.r.t. the distribution of goal sentences from the training set (L168, **R4**). Generalization is evaluated by the average success rate (SR) on goals from the testing set (**R5**). We provide a potential explanation to the surprisingly high I2C of the low coverage condition in Supp. l-251 (**R2**). We did not pair evaluations and thus did not use paired t-tests (**R2**). Substitute goals $g_s$ can belong to $\mathcal{G}^{train}$ or $\mathcal{G}^{im}$ (**R2**). Vygotsky showed that infants use more egocentric speech for planning when tasks become harder (Chap 2 - *Thought and Language*, **R2**). We agree the hierarchical algorithm presented in [36] is complementary to ours (**R4**). IMAGINE boosts low-level exploration by imagining novel goals and could be extended to a hierarchical setting (example of downstream task, **R1**). Related work on instruction following in more complicated visual environments (Das et.al 2017, Nguyen et. al 2018 and Shridhar et.al 2020) will be discussed (**R2**).

[Meta-Review · NeurIPS 2020]

Reviewers agreed that the paper proposes an interesting model for learning language conditioned goal reaching policies and performs a thorough investigation on a simple task. There was agreement that the environment studied in the paper is quite simplistic and that the paper would benefit from a task with a richer grammar/goal space. Nevertheless, the results on the task in the paper are sufficiently interesting for acceptance as a poster.